# The Flow Pattern Transition and Water Holdup of Gas–Liquid Flow in the Horizontal and Vertical Sections of a Continuous Transportation Pipe

**Guishan Ren [1], Dangke Ge [1], Peng Li [2,*], Xuemei Chen [1], Xuhui Zhang [2,3], Xiaobing Lu [2,3], Kai Sun [1], Rui Fang [1], Lifei Mi [1] and Feng Su [1]**

[1] Oil Production Technology Institute, Dagang Oilfield, Tianjin 300280, China; rengshan@petrochina.com.cn (G.R.); gedke@petrochina.com.cn (D.G.); chenxmei@petrochina.com.cn (X.C.); sunkai18@petrochina.com.cn (K.S.); fangr@petrochina.com.cn (R.F.); milfei@petrochina.com.cn (L.M.); sufeng@petrochina.com.cn (F.S.)

[2] Institute of Mechanics, Chinese Academy of Sciences, Beijing 100190, China; zhangxuhui@imech.ac.cn (X.Z.); xblu@imech.ac.cn (X.L.)

[3] School of Engineering Science, University of Chinese Academy of Sciences, Beijing 100049, China

* Correspondence: lipeng@imech.ac.cn

**Abstract:** A series of experiments were conducted to investigate the flow pattern transitions and water holdup during oil–water–gas three-phase flow considering both a horizontal section and a vertical section of a transportation pipe simultaneously. The flowing media were white mineral oil, distilled water, and air. Dimensionless numbers controlling the multiphase flow were deduced to understand the scaling law of the flow process. The oil–water–gas three-phase flow was simplified as the two-phase flow of a gas and liquid mixture. Based on the experimental data, flow pattern maps were constructed in terms of the Reynolds number and the ratio of the superficial velocity of the gas to that of the liquid mixture for different Froude numbers. The original contributions of this work are that the relationship between the transient water holdup and the changes of the flow patterns in a transportation pipe with horizontal and vertical sections is established, providing a basis for judging the flow patterns in pipes in engineering practice. A dimensionless power-law correlation for the water holdup in the vertical section is presented based on the experimental data. The correlation can provide theoretical support for the design of oil and gas transport pipelines in industrial applications.

**Keywords:** oil–water–gas flow; flow pattern; water holdup; dimensionless analysis

## 1. Introduction

The pipe transportation of oil–water–gas three-phase systems is a crucial process in oil and natural gas production and provides vital information for interpreting the production stages. A deep understanding of the flow characteristics, such as the flow patterns and water holdup (the volume fraction of water in a pipe section), of the three-phase flow is beneficial to the proper design and operation of pipelines [1]. The different flow patterns directly determine the different flow characteristics of multiphase flows. This is a notable feature of multiphase flows in pipes, and it is an essential topic in multiphase flow research. The change of the flow pattern has a vital impact on the pressure drop (reflected in the energy consumption of transportation), spatial phase distribution, and safety of pipeline transportation [2]. For example, in the churn flow of gas–liquid flow, the bubbles have different sizes and shapes, and the liquid film attached to the pipe wall becomes an up–down vibrating flow, which affects the stability of the pipeline flow. To estimate the frictional pressure gradient accurately in a transparent vertical pipe, Xu et al. [3] studied the actual flow pattern under specific flow conditions. The flow patterns can be used to deduce the concentration distribution of each phase. Jones and Zuber [4] demonstrated that the probability density function (PDF) of the fluctuations in the volume fraction could

be used as a statistical analysis tool for flow pattern identification. In short, it is crucial to predict the flow patterns and flow pattern transitions in oil pipeline transportation.

The calculation of water holdup is useful for predicting the quantity of oil in a petroleum pipeline, and water holdup is an important parameter for the classification of flow patterns. For example, Hasan and Kabir [5] proposed a semi-mechanistic method based on the flow pattern map to predict the in situ oil volume fraction and pressure drop. This model can interpret production logs to predict oil/water production rates. Liu et al. [6] developed a new annular flow model for calculating low water holdup in a horizontal pipe. Du et al. [7] experimentally investigate vertical upward oil–water two-phase flow in a 20 mm inner diameter pipe. The water holdup was measured using a vertical multiple electrode array conductance sensor, and five observed oil–water two-phase flow patterns were defined using mini-conductance probes.

Many researchers have studied the simultaneous oil–water–gas flow in a horizontal or vertical pipe. Oddie et al. [8] conducted steady-state and transient experiments of oil–water–gas multiphase flows in a large-diameter inclined pipe (11-m length, 15-cm diameter). The pipe inclination was varied from 0° (vertical) to 92°, and the flow rates of each phase were varied over wide ranges. A nuclear densitometer was used to measure the steady-state holdup values, and 10 electrical conductivities were used to provide transient and steady-state holdup profiles. The relationship between the measured holdup and flow rates, flow pattern, and pipe inclination was discussed. Spedding et al. [9] reported flow regimes for horizontal co-current oil-water-air three-phase flow for two different diameters. Combinations of dimensionless numbers for each phase were used as the mapping parameters. Two horizontal experimental three-phase facilities were used, and the flow patterns were identified using a combination of visual/video observations. Descamps et al. [10] performed laboratory experiments on the oil–water–air flow through a vertical pipe to study the gas-lifting technique for oil–water flows. The pressure gradient of the three-phase flow was always smaller than that of oil–water flow due to the air injection, except at the point of phase inversion. Air injection did not affect the concentration of oil and water at the phase inversion point. Hanafizadeh et al. [11] conducted experiments of air–water–oil three-phase flow patterns in an inclined pipe and investigated the effect of the liquid volume fraction and inclination angle on the flow patterns. The results showed that by increasing the oil cut for different inclination angles, the bubbly region was extended, and the plug region became smaller.

Numerous studies have applied computational fluid dynamics (CFD) approaches to simulate the hydrodynamics of multiphase pipe flow [12,13]. Ghorai et al. [14] developed a mathematical model to predict the holdup and pressure gradient for the water–oil–gas stratified flow in a horizontal pipe. The variations of the water holdup and pressure gradient for different situations were studied. However, the analysis was based on horizontal stratified flow only. Friedemann et al. [15] conducted a series of simulations on the gas–liquid slug flow in a horizontal concentric annulus using OpenFOAM and the built-in volume of fluid (VOF)-type solver interFoam. The simulation data were analyzed in terms of the pressure gradient and holdup profile, and they were compared to experimental data. The corresponding relationship between the flow pattern and water holdup was established. Leporini et al. [16] presented a new sand transport model implemented in one-dimensional dynamic multiphase code to deal with the liquid–solid flow as well as gas–liquid–solid flow. The numerical results demonstrated a good agreement with experimental data.

However, few studies on the comparison of the horizontal and vertical flow in a continuous transportation pipe have been reported [15,17]. In an on-site oil pipeline layout, pipelines are horizontal, vertical, and even inclined. Under the same flow parameters, pipelines in different directions may show different flow pattern characteristics, thereby creating hidden dangers for pipeline transportation stability. It is necessary to study the different flow behaviors in horizontal and vertical pipes under the same flow parameters.

The oil–water–gas flow is highly complex and related to the pipe geometry (e.g., inner pipe diameter and pipe angle), fluid properties (e.g., viscosity, density, and surface

tension), and boundary conditions (e.g., superficial input velocities) [18]. Previous studies mainly focused on the effect of a single parameter, but the coupled effect of the controlling parameters on the flow is not well understood. Hence, to understand the fundamental mechanisms of oil–water–gas flows, the controlling dimensionless parameters were derived by dimensional analysis first. In a three-phase flow, it is difficult to distinguish the boundary between oil and water, especially at high flow rates. Thus, the oil–water–gas three-phase flow can be simplified as the two-phase flow of a gas and liquid mixture considering, as the densities of oil and water are much higher than that of gas and the oil and water velocities are sufficiently high to obtain a mixture [11].

Thus, the objective of this work was to investigate the flow patterns and water holdup for a simplified gas–liquid flow in horizontal and vertical sections through a comparative study of the flow behaviors in a pipe loop considering different superficial input velocities. In particular, a series of experiments were conducted to investigate the flow pattern transition and the water holdup, considering both the horizontal and vertical sections of a transportation pipe. In addition, the relationship between the transient water holdup and the change of the flow pattern in a transportation pipe with horizontal and vertical sections was established, which provides a basis for judging the flow pattern in a pipe in engineering practice. A dimensionless power-law correlation for the water holdup in the vertical section is presented based on the experimental data.

The current paper is structured as follows. The experimental setup is described in Section 2. In Section 3, the dimensionless numbers are derived based on the physical analysis and the proper choice of the units for the problem. In Section 4, the relationship between the transient water holdup and the change of the flow pattern in a transportation pipe with horizontal and vertical sections is established, and a dimensionless power-law correlation for the water holdup in the vertical section is presented. Finally, Section 5 presents the conclusions of this study.

## 2. Description of Experiments

Figure 1 shows the schematic diagram of the oil–water–gas three-phase flow loop, which consisted of a power system, a metering system, and a mixing line. All the experiments were conducted using white mineral oil, distilled water, and air. Yellow dye was added to the oil to differentiate it from water visually. In the experiments, the temperature of the environment and the experimental section were measured by temperature sensors (Rosemount, 248 type). During the experiment, which lasted 10 h per day, the temperature of the experimental section varied from 19 °C to 22 °C. There was a long flow pattern development section between the pumps used for the fluid (water and oil) and the experimental section; so, the pumps had little effect on the fluid temperature. The changes in the physical properties of the fluid were not significant. For the sake of simplicity, the physical properties of each phase in this study were determined at atmospheric pressure and room temperature 20 °C. The physical properties of the fluids tested are presented in Table 1. The values were selected based on the experimental results of Wang [19].

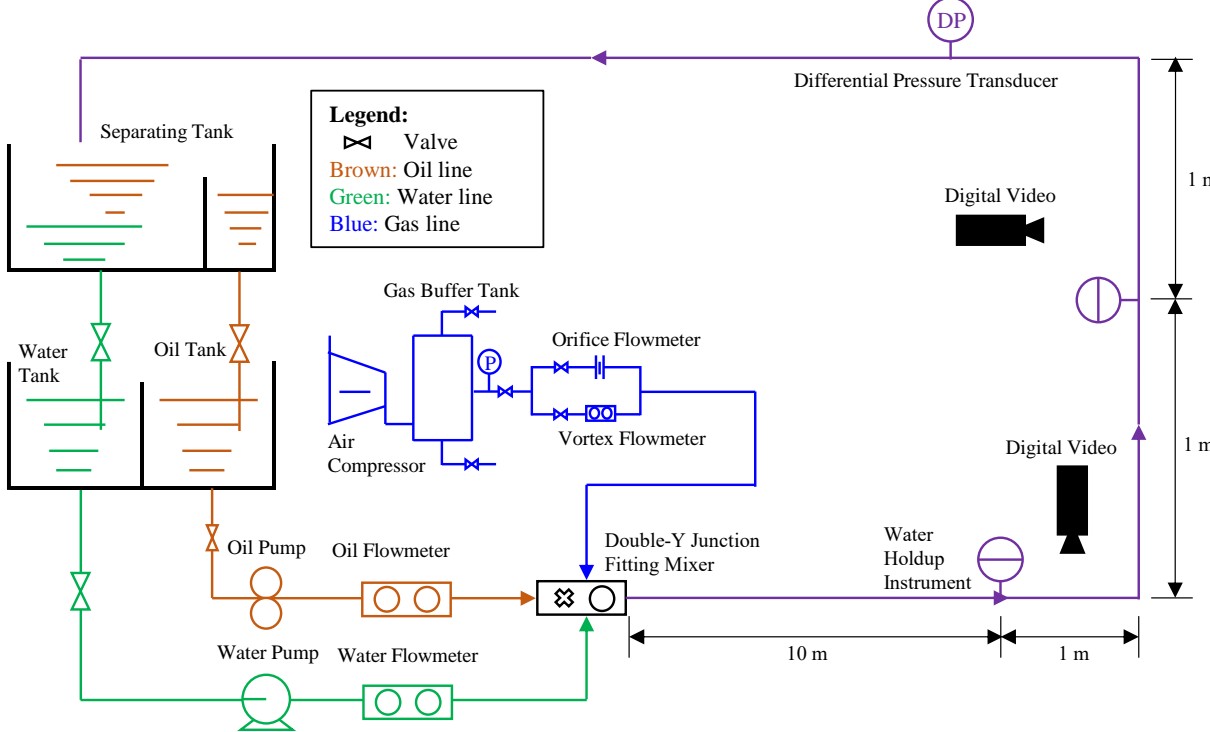

**Figure 1.** Schematic drawing of the flow loop for the oil–water–gas three-phase flow used in this study.

**Table 1.** Properties of water, oil, and gas phases measured at atmospheric pressure and 20 °C [20].

|  | Density, $\rho$ (kg/m³) | Viscosity, $\mu$ (mPa s) | Surface Tension with the Gas Phase, $\sigma$ (mN m) |
|---|---|---|---|
| White mineral oil | 841 | 28 | 30 |
| Distilled water | 998.2 | 1 | 72 |
| Air | 1.2 | 0.018 | - |

According to Wang's experimental measurements [19], the relationship between the density and temperature of white mineral oil is as follows in Equation (1):

$$\rho_o = 854.878 - 0.703t \tag{1}$$

where $\rho_0$ is the oil density, and $t$ is the temperature (°C).

The relationship between the viscosity and temperature of white mineral oil is as follows in Equation (2):

$$\mu_o = 6.075 + 67.192exp(-t/18.2) \tag{2}$$

where $\mu_o$ is the oil viscosity.

The surface tension of white mineral oil is expressed as follows in Equation (3):

$$\sigma_o = 31.77 - 0.078t \tag{3}$$

where $\sigma_o$ is the surface tension of the oil.

The surface tension of distilled water is expressed as follows in Equation (4):

$$\sigma_w = 74.33 - 0.127t \tag{4}$$

where $\sigma_w$ is the surface tension of water.

The power system pumped gas, oil, and water into the pipe. The system consisted of oil and water pumps, an oil tank, a water tank, an air compressor, and a double-Y junction fitting mixer. The metering system was composed of flowmeters and a water holdup instrument. The mixing line comprised a stainless-steel pipe section and a plexiglass pipe section with an inner diameter of 50 mm. The flow pattern development section was a horizontal stainless-steel pipe with a length of 10 m, which was convenient for the full development of the multiphase flow pattern in the pipe. The flow pattern observation section was installed at the end of the flow pattern development section, which was a plexiglass pipe, so that the flow pattern in the pipe could be easily observed. The flow observation sections were a 1-m-long horizontal transparent pipe and a 2-m-long vertical transparent pipe. Through this arrangement, the horizontal and vertical flow experiments could be carried out simultaneously.

The oil, water, and gas were pumped into the pipe from separate storage tanks. The gas was supplied by an air compressor (GA37VSDAP-13, with a capacity of 120.8 L/s) to the gas buffer tank to stabilize its pressure. The volume flow rate was regulated using an orifice flowmeter (EJA115, measurement range of 0.078–94.2 $Nm^3$/h) or a vortex flowmeter (DY015-DN15, measurement range of 30–275 $Nm^3$/h), depending on the flow range. A centrifugal pump (QABP160M2A, ABB) with a capacity of 12.5 $m^3$/h was used for the water phase, and a 6.99-KW gear pump (SNH440) with a capacity of 17 $m^3$/h and an accuracy of $\pm 0.1\%$ was used for the oil phase. The volumetric flow rates of the oil and water phases were measured by a mass flowmeter (CMF100, Micro Motion), with an accuracy of $\pm 0.1\%$. The gas, oil, and water phases entered the double-Y junction fitting mixer from the upper, middle, and lower layers of the mixer, respectively. A schematic diagram of the double-Y junction fitting mixer is shown in Figure 2. The well-mixed three-phase flow passed through the test section and then flowed back to the separating tank, in which the gas escaped to the atmosphere and the oil and water flowed into the oil and water tanks, respectively.

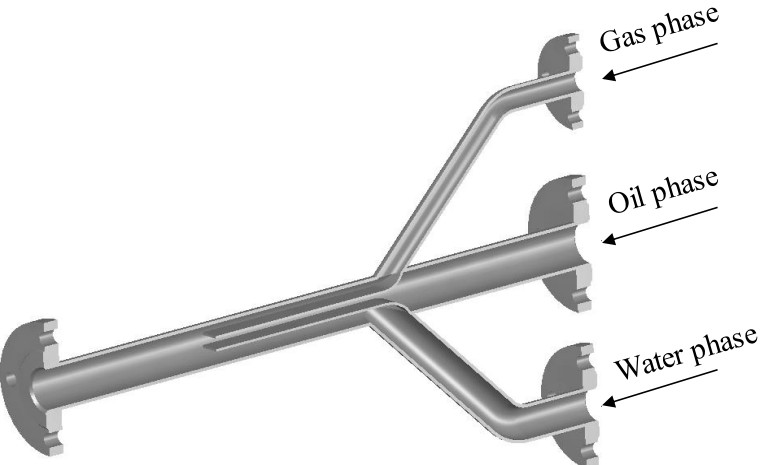

**Figure 2.** Schematic of the double-Y junction fitting mixer.

When the pressure drop in the pipe remained constant, it was deemed that a steady state of the system had been reached. Data for a period of 300 s were recorded. The pressure drop was measured by a differential pressure transducer placed in the return section. A digital video was used for flow pattern identification. The cross-sectional average water holdup $\alpha_w$ (in situ volume fraction of water) was recorded by two water holdup instruments placed in the horizontal and vertical sections of the test pipe. The water holdup instrument was equipped with a conductance probe having a sampling frequency of 1 Hz. A photograph of the instrument is shown in Figure 3. Two pairs of conductance probes were regularly distributed in the middle of the stainless steel pipe. Each probe was comprised of two parallel brass rods. When alternating current flowed

between two probes, the conductance probe measured the voltage between the two ends of the conductor, which reflects the mean conductivity of the mixture in the pipe [20]. Because the conductivities of oil and gas are weak, the voltage values measured when the pipe was filled with pure oil or gas were basically the same. Calibration of the water holdup instrument was performed by measuring the transmitted conductivity for single-phase gas, oil, and water. The calibration information was used to calculate the water holdup for three-phase flows. The measured voltage values are denoted by $V_o$, $V_w$, and $V_g$ when the pipe is fully filled with the pure oil, water, and gas, respectively.

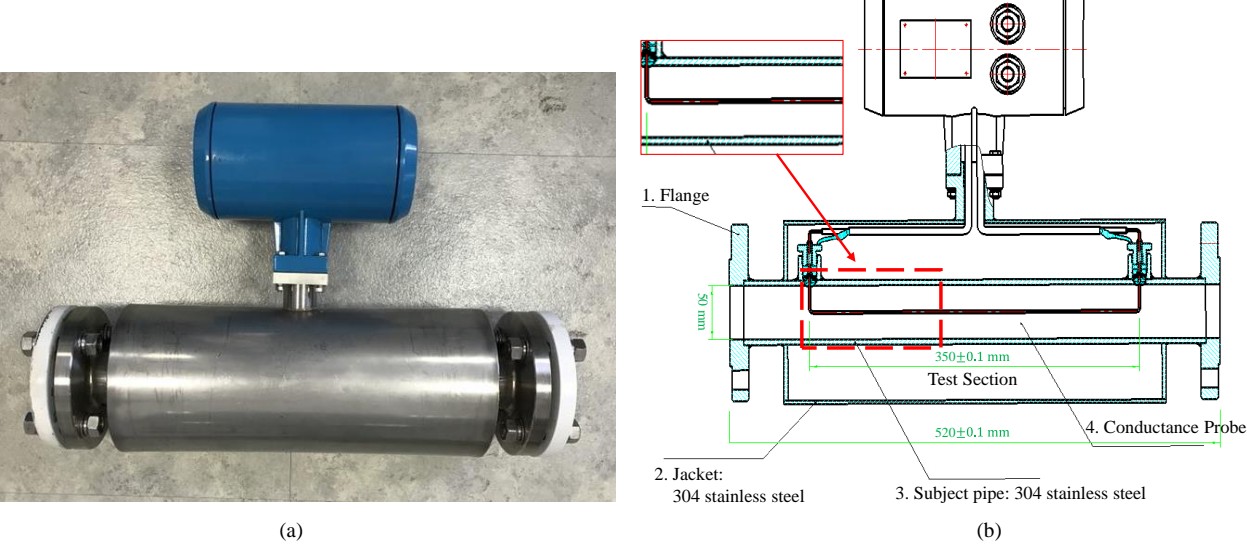

(a)                                     (b)

**Figure 3.** (**a**) Photograph of the water holdup instrument designed for the experiments. (**b**) The internal structure of the water holdup instrument.

We inferred that the mean voltage $V_{exp}$ measured across the flow stream would be a function of the form (Equation (5)):

$$V_{exp} = f(\alpha_w, \alpha_o, V_w, V_o, V_g) \tag{5}$$

The volume fraction of the three phases satisfies the following relation (Equation (6)):

$$\alpha_o + \alpha_w + \alpha_g = 1 \tag{6}$$

Equation (5) can be made dimensionless by using $V_w$ as the characteristic voltage, yielding the following (Equation (7)):

$$\frac{V_{exp}}{V_w} = f(\alpha_w, \alpha_o, \frac{V_0}{V_w}, \frac{V_g}{V_w}) \tag{7}$$

As a classic method, the conductivity measurement method has been widely studied and applied due to its simple structure, convenient installation, and fast response. However, the equivalent conductivity of multiphase fluid is not only related to phase holdup but also affected by flow pattern or phase distribution [21]. This issue has not been solved for more complex flow patterns since the first work by Bruggeman [22]. In general, the mean voltage $V_{exp}$ measured across the flow stream can be assumed to be a linear superposition of the corresponding parameters of each phase for simplicity. Achwal and Stepanek [23,24] used a conductance probe to measure the liquid holdup in a gas–liquid system. As they pointed out, the electroconductivity of a liquid system is proportional to the cross-sectional area of the conducting liquid. Thus, the conductivity should be proportional to the liquid holdup. The same procedure was also adopted by Begovich and Watson [25]. Du et al. [7]

measured the water holdup using a vertical multiple electrode array conductance sensor in vertical upward oil–water flow. According to the experiments, the mean voltage for mixed fluid and oil holdup showed a good linear relationship. Based on the above considerations, the mean voltage $V_{exp}$ can be expressed as follows in Equation (8):

$$\frac{V_{exp}}{V_w} = \alpha_0 \frac{V_0}{V_w} + (1 - \alpha_w - \alpha_o)\frac{V_g}{V_w} + \alpha_w \tag{8}$$

Since the voltage $V_o$ is equal to $V_g$, the above formula can be further simplified as follows in Equation (9):

$$\frac{V_{exp}}{V_w} = (1 - \alpha_w)\frac{V_o}{V_w} + \alpha_w \tag{9}$$

Thus, the relationship between the water holdup $\alpha_w$ and the mean voltage $V_{exp}$ can be expressed as follows:

$$\alpha_w = \frac{V_0 - V_{exp}}{V_o - V_w} \tag{10}$$

The measured voltage values when the pipe is fully filled with the pure water and pure oil in each group of experiments were different. Therefore, $V_o$ and $V_w$ were measured for each set of experimental conditions when the voltage signal was converted into a water holdup using Equation (10). The dimensionless electrical signal of the water holdup instrument ensured that the measured data of the horizontal and vertical devices in the same group of experiments could be compared, which was also convenient for the comparison of experimental data of different groups.

### 3. Analysis of Oil–Water–Gas Three-Phase Flow

An oil–water–gas three-phase flow is highly complex and related to pipe geometry, fluid properties, and fluid flow rates. An oil–water–gas three-phase flow can be regarded as a special kind of gas–liquid two-phase flow, especially at high flow rates, where the oil and water are well mixed and form a homogeneous dispersion. The clear identification of the oil and the water phases is difficult in these cases [8]. The methods and theories developed for gas–liquid two-phase flows can be used as the basis for the investigation of oil–water–gas three-phase flows [14]. The liquid mixture properties, such as the viscosity and density, depend on the ratio of the superficial velocity of oil to that of water. The mixture density is defined as follows in Equation (11):

$$\rho_m = \varepsilon_w \rho_w + \varepsilon_o \rho_o \tag{11}$$

where $\rho_w$ is the water density. $\varepsilon_w$ and $\varepsilon_o$ are the input water and oil cuts, respectively, defined as the ratio of each phase flow rate to the mixture flow rate (Equations (12) and (13)):

$$\varepsilon_w = \frac{Q_w}{Q_w + Q_o} = \frac{u_{sw}}{u_{sm}} \tag{12}$$

$$\varepsilon_o = \frac{Q_o}{Q_w + Q_o} = \frac{u_{so}}{u_{sm}} \tag{13}$$

where $Q$ is the volume flow rate of each phase, and $u_s$ is the superficial velocity, which is defined as $u_s = Q/A$, $A$ is the cross-sectional area of the pipe, $A = \pi D^2/4$, and $D$ is the inner pipe diameter.

An oil–water mixture is a non-Newtonian fluid, and its viscosity is called the apparent viscosity. In general, the viscosity of a liquid mixture varies significantly with the spatial distribution of the two phases, the viscosity of each phase, the temperature, and the pressure. It is difficult to include all these factors in any theoretical expression of the apparent viscosity. Some scholars have suggested using a calculation method similar to that for the densities of oil–water mixtures to calculate the apparent viscosity [11,26,27]. Over the range of superficial velocities considered here, the oil and water were well mixed,

and the liquid phases appeared "milky." In this study, it was assumed that the liquid mixture viscosity depended on the water and oil fractions for convenience. The mixture viscosity is given as follows in Equation (14):

$$\mu_m = \varepsilon_w \mu_w + \varepsilon_o \mu_o \tag{14}$$

where $\mu_m$ is the water viscosity.

Dimensional analysis is a useful tool to obtain the coupling effect of the controlling factors on the two-phase flow behavior. The factors affecting the gas–liquid two-phase flow are listed as follows:

- Gas phase: density $\rho_g$, viscosity $\mu_g$.
- Liquid mixture: density $\rho_m$, viscosity $\mu_m$, interfacial tension $\sigma$.
- Geometric parameter: inner pipe diameter $D$.
- Boundary condition: superficial velocity of gas $u_{sg}$, superficial velocity of the liquid mixture $u_{sm}$.
- Gravitational acceleration: $g$.

The final steady-state of the gas–liquid flow system, characterized by the water holdup $\alpha_w$ and the flow pattern, is a function of the above control parameters (Equation (15)):

$$\begin{cases} \alpha_w = f(\rho_g, \mu_g, u_{sg}, \sigma, \rho_m, \mu_m, u_{sm}; D, g), \ \varepsilon_w \neq 0 \\ flow\ pattern = f(\rho_g, \mu_g, u_{sg}, \sigma, \rho_m, \mu_m, u_{sm}; D, g) \end{cases} \tag{15}$$

The above formula can be nondimensionalized as follows in Equation (16):

$$\begin{cases} \alpha_w = f\left( \dfrac{u_{sg}}{u_{sm}}, \dfrac{\rho_m u_{sm} D}{\mu_m}, \dfrac{u_{sm}^2}{gD}, \dfrac{|\rho_m - \rho_g| g D^2}{\sigma}, \dfrac{\mu_g}{\mu_m}, \dfrac{\rho_g}{\rho_m} \right), \ \varepsilon_w \neq 0 \\ flow\ pattern = f\left( \dfrac{u_{sg}}{u_{sm}}, \dfrac{\rho_m u_{sm} D}{\mu_m}, \dfrac{u_{sm}^2}{gD}, \dfrac{|\rho_m - \rho_g| g D^2}{\sigma}, \dfrac{\mu_g}{\mu_m}, \dfrac{\rho_g}{\rho_m} \right) \end{cases} \tag{16}$$

where $u_{sg}/u_{sm}$ is the gas-to-liquid superficial velocity ratio, $\rho_m u_{sm} D / \mu_m$ is the Reynolds number $Re_m$, $u_{sm}^2 / gD$ is the Froude number $Fr_m$, $(|\rho_m - \rho_g| g D^2)/\sigma$ is the Eötvös number $Eo$, which represents the ratio of the buoyancy force to the surface tension force, $\mu_g / \mu_m$ is the viscosity ratio, and $\rho_g / \rho_m$ is the density ratio. In this study, the input water cut $\varepsilon_w$ ranged from 0% to 100%, corresponding to $\mu_m$ values from 1 to 32 mPa/s and $\rho_m$ values from 843 to 998.2 kg/m3. Because the density and viscosity of the gas were much lower than that of the liquid mixture, the effect of variations of $Eo$, $\rho_g / \rho_m$, and $\mu_g / \mu_m$ were not considered in this study. Thus, Equation (16) can be simplified as follows:

$$\begin{cases} \alpha_w = f\left( \dfrac{u_{sg}}{u_{sm}}, Re_m, Fr_m \right), \ \varepsilon_w \neq 0 \\ flow\ pattern = f\left( \dfrac{u_{sg}}{u_{sm}}, Re_m, Fr_m \right) \end{cases} \tag{17}$$

The input water cut $\varepsilon_w$ is an important parameter for oil pipeline transportation. The effect of $\varepsilon_w$ on the final steady-state is reflected in the Reynolds number $Re_m$ Tests were conducted for different values of the dimensionless parameters in Equation (17) to associate the observed flow patterns with the measured water holdup values for the horizontal and vertical sections of the pipe. All the experimental values of $u_{sg}$, $u_{sg}$, and $\varepsilon_w$ and the corresponding dimensionless numbers are given in Table 2.

**Table 2.** Parameter values of flow conditions imposed during the oil–water–gas pipe flow experiments.

| Test | $\varepsilon_w$ | $u_{sg}/u_{sm}$ | $Re_m = \rho_m u_{sm} D / u_{sm}$ | $Fr_m = u_{sm}^2/gD$ |
|------|-----|-----------------|-----------------------------------|----------------------|
| 1 | 1.0 | 0.25, 0.50, 1, 2, 3, 4, 5, 6 | 11,303 | 0.105 |
| 2 | 0.9 | 0.25, 0.50, 1, 2, 3, 4, 5, 6 | 2714 | 0.105 |
| 3 | 0.8 | 0.25, 0.50, 1, 2, 3, 4, 5, 6 | 1521 | 0.105 |
| 4 | 0.7 | 0.25, 0.50, 1, 2, 3, 4, 5, 6 | 1046 | 0.105 |
| 5 | 0.6 | 0.25, 0.50, 1, 2, 3, 4, 5, 6 | 791 | 0.105 |
| 6 | 0.5 | 0.25, 0.50, 1, 2, 3, 4, 5, 6 | 632 | 0.105 |
| 7 | 0.4 | 0.25, 0.50, 1, 2, 3, 4, 5, 6 | 523 | 0.105 |
| 8 | 0.3 | 0.25, 0.50, 1, 2, 3, 4, 5, 6 | 444 | 0.105 |
| 9 | 0.2 | 0.25, 0.50, 1, 2, 3, 4, 5, 6 | 384 | 0.105 |
| 10 | 0.1 | 0.25, 0.50, 1, 2, 3, 4, 5, 6 | 336 | 0.105 |
| 11 | 0.0 | 0.25, 0.50, 1, 2, 3, 4, 5, 6 | 298 | 0.105 |
| 12 | 1.0 | 0.25, 0.50, 1, 2, 3, 4, 5, 6 | 16,248 | 0.216 |
| 13 | 0.9 | 0.25, 0.50, 1, 2, 3, 4, 5, 6 | 3901 | 0.216 |
| 14 | 0.8 | 1, 2, 3, 4, 5, 6 | 2187 | 0.216 |
| 15 | 0.7 | 0.25, 0.50, 1, 2, 3, 4, 5, 6 | 1504 | 0.216 |
| 16 | 0.6 | 0.25, 0.50, 1, 2, 3, 4, 5, 6 | 1137 | 0.216 |
| 17 | 0.5 | 0.25, 0.50, 1, 2, 3, 4, 5, 6 | 908 | 0.216 |
| 18 | 0.4 | 0.25, 0.50, 1, 2, 3, 4, 5, 6 | 752 | 0.216 |
| 19 | 0.3 | 0.25, 0.50, 1, 2, 3, 4, 5, 6 | 638 | 0.216 |
| 20 | 0.2 | 0.25, 0.50, 1, 2, 3, 4, 5, 6 | 551 | 0.216 |
| 21 | 0.1 | 0.25, 0.50, 1, 2, 3, 4, 5, 6 | 484 | 0.216 |
| 22 | 0.0 | 0.25, 0.50, 1, 2, 3, 4, 5, 6 | 429 | 0.216 |
| 23 | 1.0 | 0.25, 0.50, 1, 2, 3, 4 | 21,193 | 0.368 |
| 24 | 0.9 | 0.25, 0.50, 1, 2, 3, 4 | 5089 | 0.368 |
| 25 | 0.8 | 0.25, 0.50, 1, 2, 3, 4 | 2852 | 0.368 |
| 26 | 0.7 | 0.25, 0.50, 1, 2, 3, 4 | 1962 | 0.368 |
| 27 | 0.6 | 0.25, 0.50, 1, 2, 3, 4 | 1483 | 0.368 |
| 28 | 0.5 | 0.25, 0.50, 1, 2, 3, 4 | 1185 | 0.368 |
| 29 | 0.4 | 0.25, 0.50, 1, 2, 3, 4 | 980 | 0.368 |
| 30 | 0.3 | 0.25, 0.50, 1, 2, 3, 4 | 832 | 0.368 |
| 31 | 0.2 | 0.25, 0.50, 1, 2, 3, 4 | 719 | 0.368 |
| 32 | 0.1 | 0.25, 0.50, 1, 2, 3, 4 | 631 | 0.368 |
| 33 | 0.0 | 0.25, 0.50, 1, 2, 3, 4 | 559 | 0.368 |

## 4. Results and Discussion

### 4.1. Flow Pattern Maps

The identification of the flow patterns was based on both visual observations from the video camera and the PDF of the instantaneous cross-sectional water holdup measured by the water holdup instrument. According to the classification by Weisman [28], the flow patterns for a gas–liquid flow were classified into bubbly, slug, plug, annular, stratified, and disperse flows in the horizontal pipe and bubbly, slug, churn, annular, and disperse flows in the vertical pipe. Figure 4 shows the schematic representations of the horizontal and vertical gas–liquid flow patterns defined by Weisman [28]. Over the range of superficial velocities considered here, plug, slug, and annular flows were observed in the horizontal section. In the vertical section, slug and churn flows were observed.

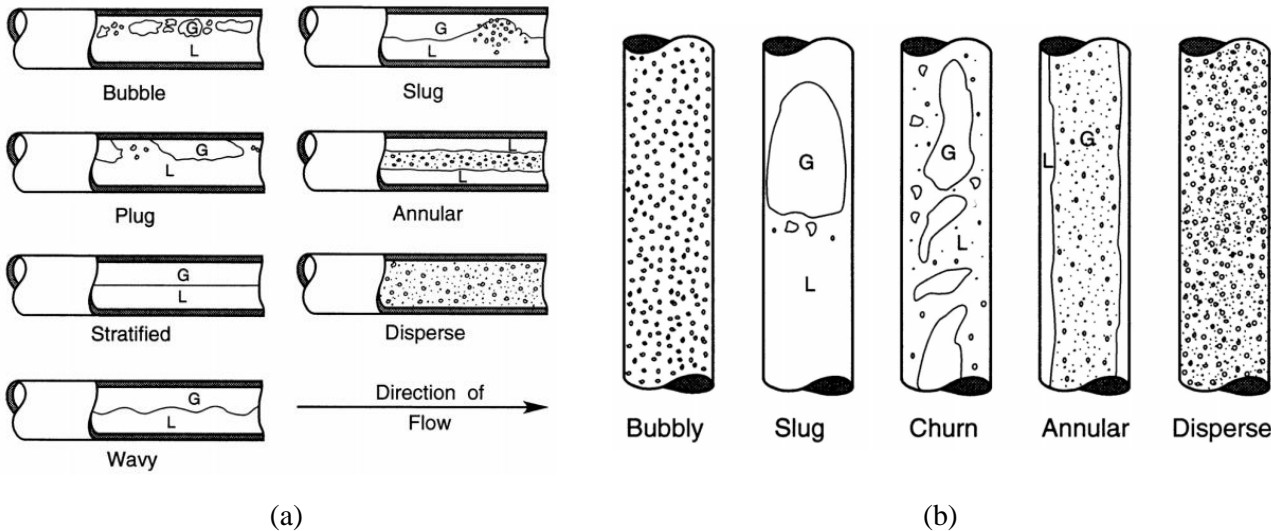

(a)　　　　　　　　　　　　　　　　　　　　　　　　　(b)

**Figure 4.** Schematic representation of the (**a**) horizontal and (**b**) vertical gas–liquid flow patterns defined by Weisman [28].

Figure 5 shows examples of the four main flow patterns that were observed in this study in both the horizontal and vertical sections in a continuous transportation pipe. Figure 5a presents the plug flow pattern, where small bubbles gathered in the upper part of the pipe and formed large bubbles, and there were almost no small bubbles between the large bubbles. Figure 5b shows the annular flow pattern, where a liquid film with a certain thickness formed between the gas column and the pipe wall. Figure 5c shows the slug flow pattern, where gas pockets were separated by slugs of liquid with dispersed small bubbles. Figure 5d shows the churn flow pattern; the flow was similar to the slug flow but without clear phase separation or structure. Figure 5 shows that the oil and water phases were well mixed with each other. Based on the experimental observations, the assumption that the oil–water phase was simplified as a liquid mixture was reasonable.

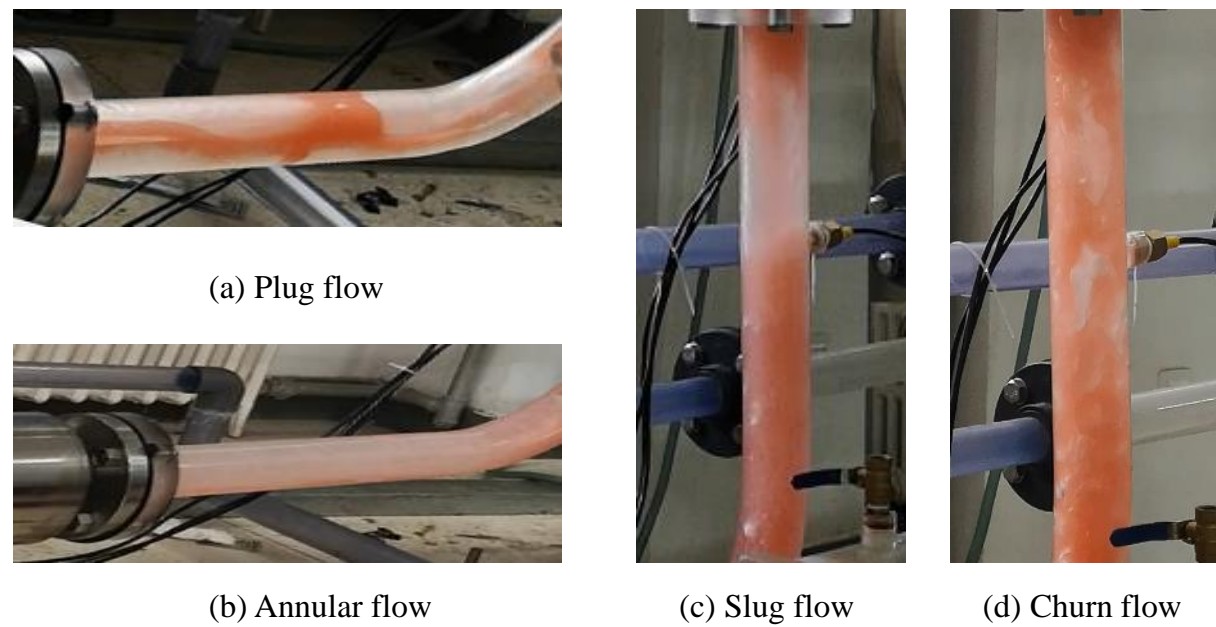

(a) Plug flow

(b) Annular flow　　　　　　　　(c) Slug flow　　　　　　(d) Churn flow

**Figure 5.** Instantaneous flow patterns observed in the experiments in the horizontal and vertical sections: (**a**) plug flow, $Fr_m = 0.368$, $u_{sg}/u_{sm} = 0.25$, $\varepsilon_w = 0.5$; (**b**) annular flow $Fr_m = 0.105$, $u_{sg}/u_{sm} = 0.50$, $\varepsilon_w = 0.5$; (**c**) slug flow, $Fr_m = 0.216$, $u_{sg}/u_{sm} = 6$, $\varepsilon_w = 0.1$; (**d**) churn flow, $Fr_m = 0.368$, $u_{sg}/u_{sm} = 4$, $\varepsilon_w = 0.1$.

Figures 6 and 7 show the typical time-series data of the cross-sectional average water holdup $\alpha_w$ and the corresponding PDF as a function of the water holdup for different superficial velocities in both the horizontal and vertical sections [4]. Figure 6 shows the data for $Fr_m = 0.368$, $u_{sg}/u_{sm} = 0.50$, and $\varepsilon_w = 0.8$. In the horizontal section, plug flow was observed. With the increase in the gas flow rate, small bubbles coalesced in the upper part of the pipe to form large bubbles. Few small bubbles existed between the large bubbles. The plug flows were characterized by two peaks in the PDF at $\alpha_w = 0.55$ and 0.75 in this case. The two peaks corresponded to the bubble region and the plug region, respectively. In the vertical section, slug flow was observed under the same experimental conditions. Taylor bubbles and liquid slugs between two adjacent Taylor bubbles formed slug flows. Many small bubbles were distributed between the Taylor bubbles, which is referred to as the wake region [29]. In this study, slug flows were characterized by three peaks in the PDF at $\alpha_w = 0.45$, 0.60, and 0.80. The regions at $\alpha_w = 0.45$, 0.60, and 0.80 corresponded to the Taylor bubble, wake, and liquid slug regions, respectively.

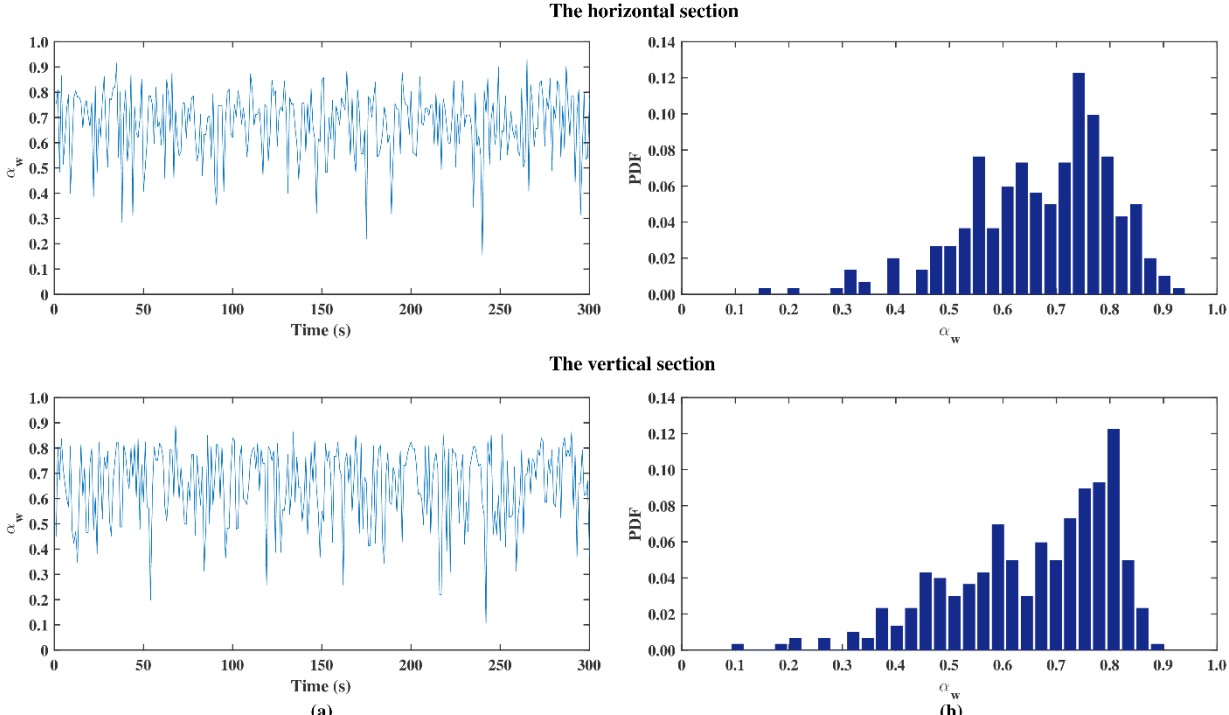

**Figure 6.** Typical evolution of the cross-sectional average water holdup $\alpha_w$ and the probability density function (PDF) of the fluctuations in the volume fraction for oil–water–gas pipe flow ( $Fr_m = 0.368$, $u_{sg}/u_{sm} = 0.50$, and $\varepsilon_w = 0.8$). Plug flow in the (**a**) horizontal section and slug flow in the (**b**) vertical section.

Figure 7 shows the result for $Fr_m = 0.105$, $u_{sg}/u_{sm} = 6$, and $\varepsilon_w = 0.5$. Annular flow was observed in the horizontal section in this case. The PDF showed a single narrow peak at a low input water holdup ($\alpha_w = 0.02$). The point at which the maximum occurred in the annular flow PDF represents the volume fraction of the liquid film surrounding the gas column. In the vertical section, churn flow was observed, and the associated PDF diagram showed a single peak that was similar to the PDF associated with the annular flow at a low input water cut. However, the distribution range of the single peak was greater than that in the annular flow, suggesting that significantly severely agitated mixing occurred.

Flow pattern maps were developed for the oil–water–gas pipe flow in the horizontal and vertical sections, which are shown in Figures 8 and 9. In these figures, the x-, y-, and z-axes corresponded to the dimensionless numbers $Fr_m$, $u_{sg}/u_{sm}$, and $Re_m$, respectively. $Fr_m$ values of 0.105, 0.216, and 0.368 were investigated, and $Re_m$ and $u_{sg}/u_{sm}$ were in the ranges of 298–21,193 and 0.25–6, respectively. The flow pattern maps depended on the pipe size

and the size of the gas inclusions. The flow pattern maps in this study were constructed for $D = 50$ mm.

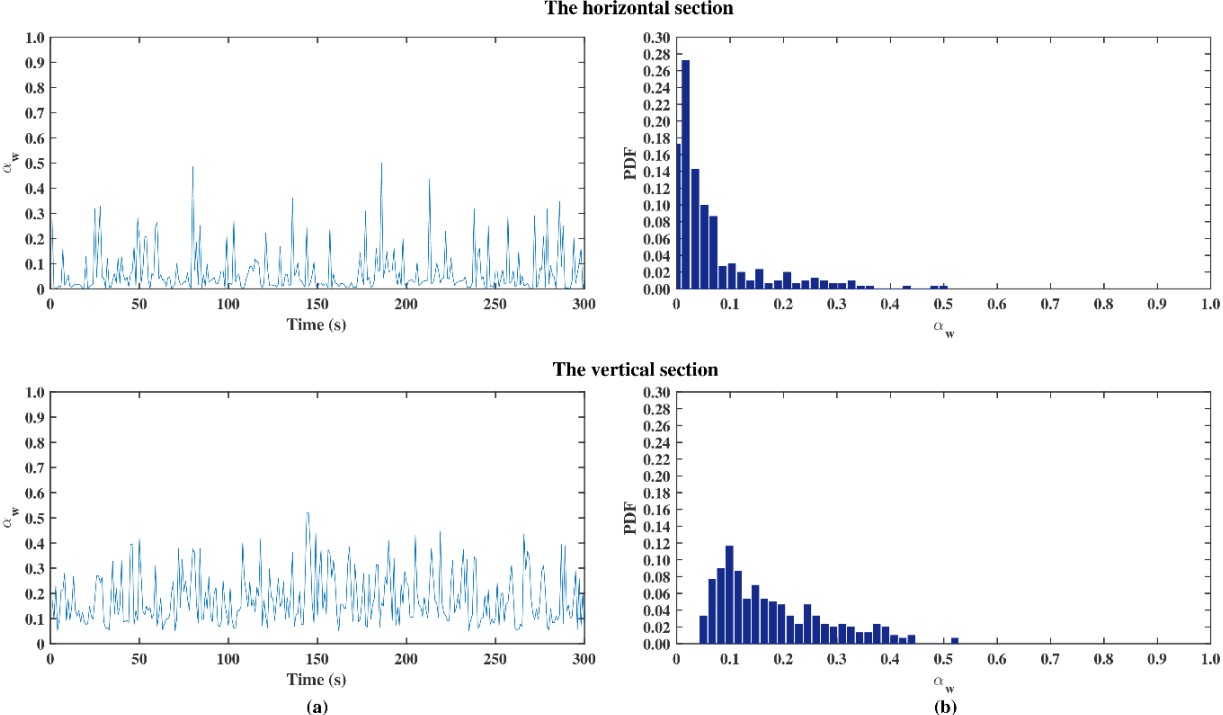

**Figure 7.** Typical evolution of the cross-sectional average water holdup $\alpha_w$ and the probability density function (PDF) of the fluctuations in the volume fraction for oil–water–gas pipe flow ($Fr_m = 0.105$, $u_{sg}/u_{sm} = 6$, and $\varepsilon_w = 0.5$). Annular flow in the (**a**) horizontal section and churn flow in the (**b**) vertical section.

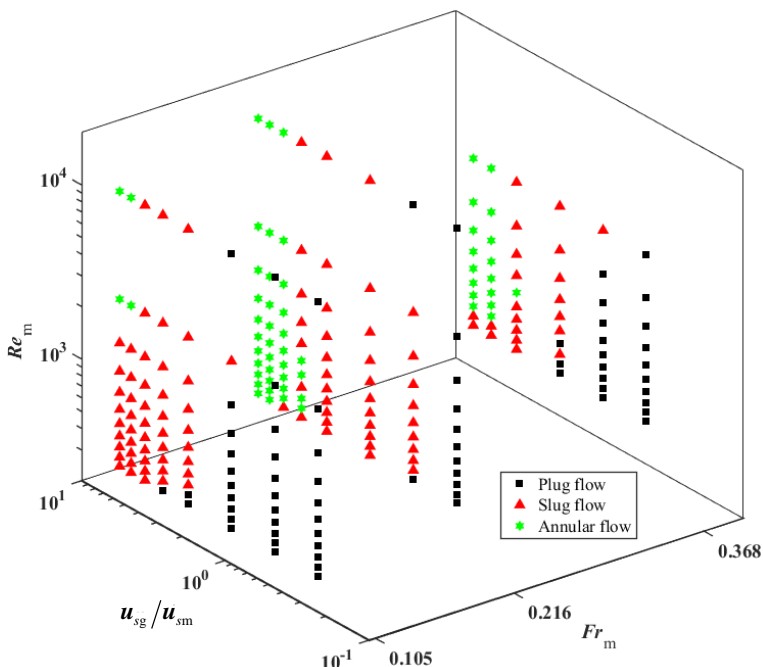

**Figure 8.** Flow pattern maps in the horizontal section, in which the dimensionless number $Re_m$ ranged from 298 to 21,193, the dimensionless number $u_{sg}/u_{sm}$ ranged from 0.25 to 6, and the dimensionless number $Fr_m$ was fixed at 0.105, 0.216, and 0.368.

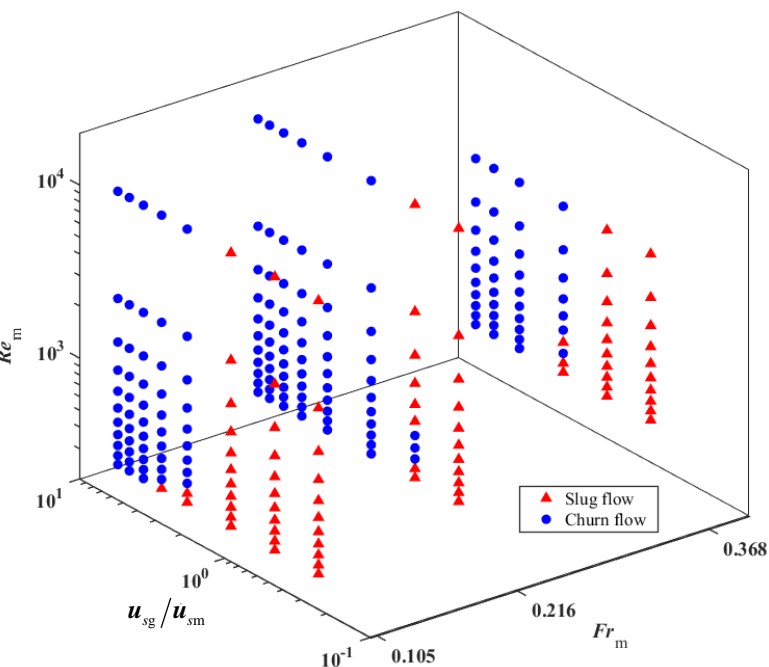

**Figure 9.** Flow pattern maps in the vertical section, in which the dimensionless number $Re_m$ ranged from 298 to 21,193, the dimensionless number $u_{sg}/u_{sm}$ ranged from 0.25 to 6, and the dimensionless number $Fr_m$ was fixed at 0.105, 0.216, and 0.368.

In the horizontal section (Figure 8), at low $u_{sg}/u_{sm}$ values, plug flow was observed. For a given $Re_m$ and increasing $u_{sg}/u_{sm}$, the gas content increased, converting plug flow to slug or annular flow. At high $Re_m$ and $u_{sg}/u_{sm}$, annular flow was identified. By increasing $Fr_m$ from 0.105 to 0.216 and 0.368, the transition boundary between the slug and annular flows moved to the right side of the map. In other words, by increasing $Fr_m$, the annular flow regime zone was expanded. The experimental flow pattern map in the vertical section is shown in Figure 9. Only slug and churn flows were observed in the experimental conditions. A further increase in $u_{sg}/u_{sm}$ converted the slug flow to churn flow. Compared with the horizontal section, plug flow was replaced by slug flow at low $u_{sg}/u_{sm}$ values, and slug flow was replaced by churn flow at high $u_{sg}/u_{sm}$ values. By increasing $Fr_m$, the transition boundary between the slug and churn flows moved to the right side of the map, i.e., lower $u_{sg}/u_{sm}$ values.

The comparison of Figures 8 and 9 shows that the plug flow in the horizontal section corresponded to the slug flow in the vertical section, and the slug flow in the horizontal section corresponded to the churn flow in the vertical section. At low $u_{sg}/u_{sm}$ values, the vertical section exhibited slug flow, while in the horizontal section, bubbles could float to the upper part of the pipe to form plug flow due to buoyancy. At high $u_{sg}/u_{sm}$ values, the horizontal section exhibited slug flow due to the inertial forces of the bubbles being very large, while in the vertical section, the bubbles broke up and formed churn flow instead.

*4.2. Water Holdup*

In this study, the water holdup instrument was employed to measure the steady-state water holdup in the horizontal and vertical sections of the pipe. However, the measured holdup data for the horizontal section at high $u_{sg}/u_{sm}$ values were considered to be less reliable because the conductance probe was placed parallel to the flow direction. Therefore, only the water holdup measured in the vertical section is considered in the following discussion. In the following work, the structure of the water holdup instrument can be improved to get more accurate experimental data in the horizontal section.

Figure 10 illustrates the relationship between the measured water holdup $\alpha_w$ and $u_{sg}/u_{sm}$ for different $Re_m$ values. $Fr_m$ values of 0.105, 0.216, and 0.368 were investigated.

The increase in $u_{sg}/u_{sm}$ led to a decrease in the water holdup. For a given $u_{sg}/u_{sm}$, the water holdup decreased with the decrease in $Re_m$. Moreover, for large $u_{sg}/u_{sm}$, the amount of decrease of the water holdup became smaller and smaller until it remained unchanged. For low $Re_m$ (corresponding to input water cuts of 0.1 and 0.2), the error of the measured water holdup was large, causing the water holdup to show different trends. According to Figure 10a–c, it is noted that the trend between $\alpha_w$ and $u_{sg}/u_{sm}$ remained essentially the same, regardless of the value of $Fr_m$. The value of $u_{sg}/u_{sm}$ had an important influence on the water holdup $\alpha_w$.

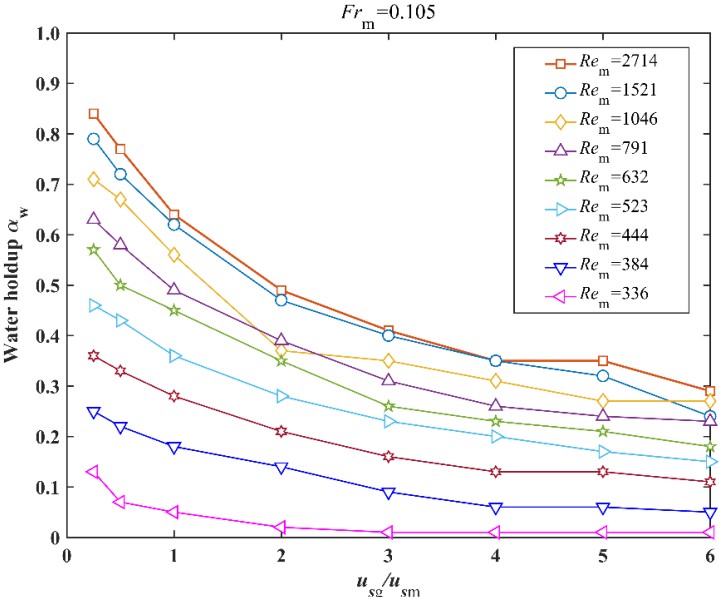

(**a**) Water holdup $\alpha_w$ versus $u_{sg}/u_{sm}$ for $Fr_m$ = 0.105

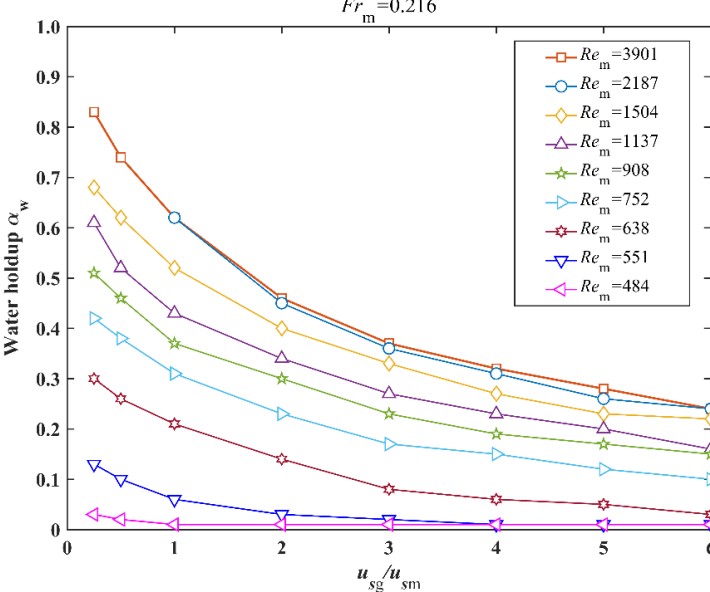

(**b**) Water holdup $\alpha_w$ versus $u_{sg}/u_{sm}$ for $Fr_m$ = 0.216

**Figure 10.** *Cont.*

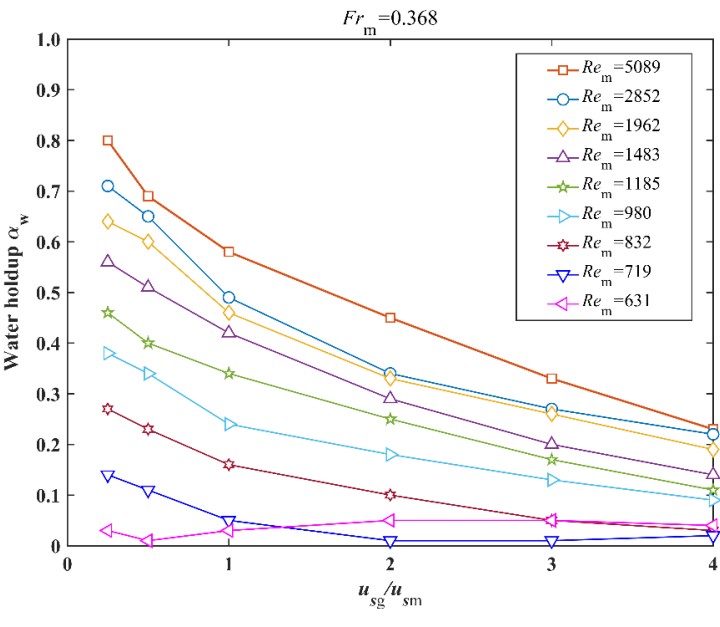

(**c**) Water holdup $\alpha_w$ verses $u_{sg}/u_{sm}$ for $Fr_m$ = 0.368

**Figure 10.** Effect of $u_{sg}/u_{sm}$ on water holdup $\alpha_w$ in the vertical section for different $Re_m$ values. The input water cut $\varepsilon_w$ ranged from 10 to 90%. $Fr_m$ was fixed at (**a**) 0.105, (**b**) 0.216, and (**c**) 0.368.

Figure 11 shows the water holdup $\alpha_w$ as a function of $Re_m$ for different $u_{sg}/u_{sm}$. The data for three different $Fr_m$ values (0.105, 0.216, and 0.368) are presented. The increase in $Re_m$ led to an increase in the water holdup. For a given $Re_m$, the water holdup decreased with the increase in $u_{sg}/u_{sm}$. The results were reasonably close to the results shown in Figure 10 and exhibited similar trends. By analogy to Figure 11a–c, it is noted that the trend between $\alpha_w$ and $Re_m$ remained essentially the same, regardless of the values of $Fr_m$. The Reynolds number $Re_m$ is also a key parameter affecting the flow behavior of the oil–water–gas three-phase flow.

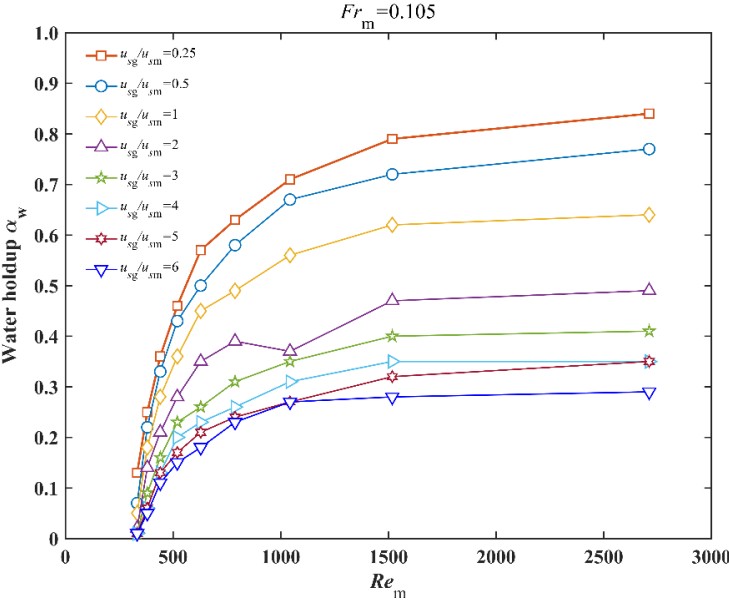

(**a**) Water holdup $\alpha_w$ verses $Re_m$ for $Fr_m$ = 0.105

**Figure 11.** *Cont.*

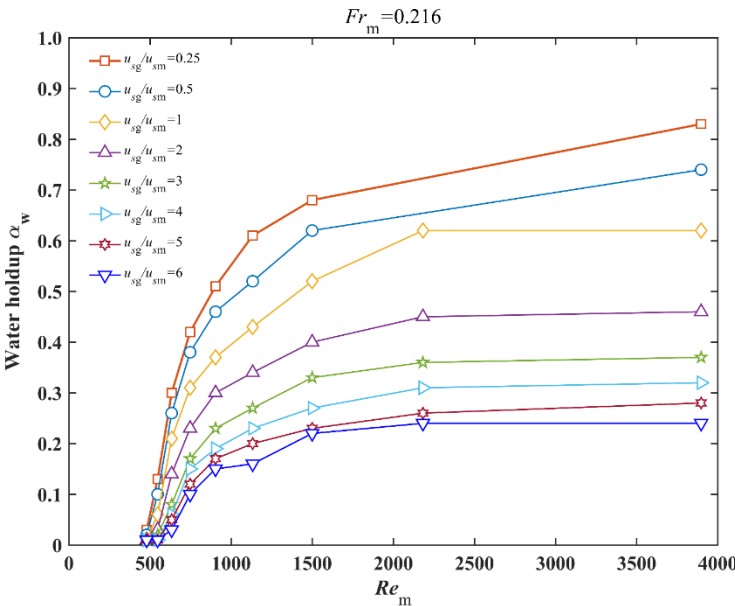

(**b**) Water holdup $\alpha_w$ verses $Re_m$ for $Fr_m$ = 0.216

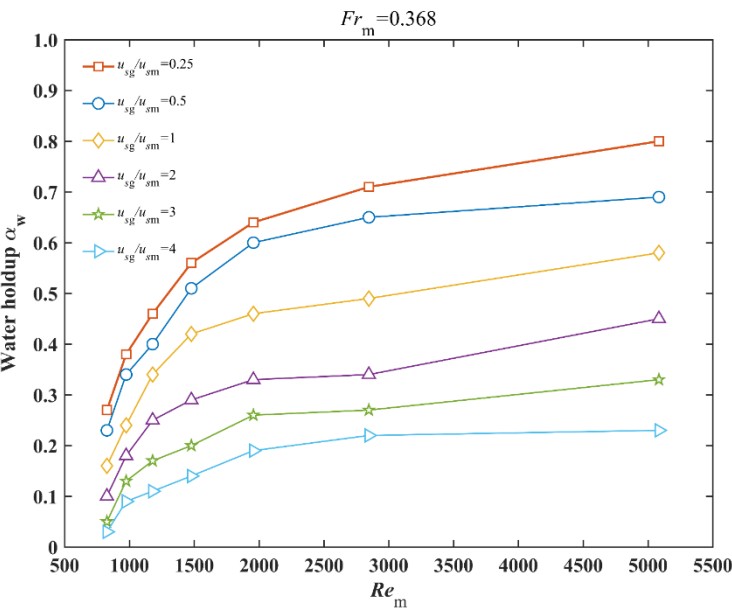

(**c**) Water holdup $\alpha_w$ verses $Re_m$ for $Fr_m$ = 0.368

**Figure 11.** Effect of $Re_m$ on water holdup $\alpha_w$ in the vertical section for different $u_{sg}/u_{sm}$ values. The input water cut $\varepsilon_w$ ranged from 10 to 90%. $Fr_m$ was fixed at (**a**) 0.105, (**b**) 0.216, and (**c**) 0.368.

According to Equation (17), the water holdup in the vertical section $\alpha_w$ is a function of $Re_m$, $Fr_m$, and $u_{sg}/u_{sm}$. The relationship between $\alpha_w$ and the corresponding dimensionless numbers can be written as the following power-law correlation in Equation (18):

$$\alpha_w = a(Re_m)^b(Fr_m)^c \left( \frac{u_{sg}}{u_{sm}} \right)^d, \ \varepsilon_w \neq 0 \tag{18}$$

where $a$ is a pre-factor, and $b$, $c$, and $d$ are the fitted exponents. The experimental data were fitted to obtain the best correlation through the least-squares method. The fitting result is as follows:

$$\alpha_w = 0.06(Re_m)^{0.20}(Fr_m)^{-0.26} \left( \frac{u_{sg}}{u_{sm}} \right)^{-0.34}, \ \varepsilon_w \neq 0 \tag{19}$$

i.e., $a = 0.06$, $b = 0.20$, $c = -0.26$, and $d = -0.34$. The data from which the above correlation was derived were collected for the following parameter ranges: $0.25 \leq u_{sg}/u_{sm} \leq 6$, $336 \leq Re_m \leq 5089$, and $0.105 \leq Fr_m \leq 0.368$. The coefficient of determination $R^2$ of Equation (19) was 0.83. The standard deviation (SD) of the predicted value was 8.5%, which was calculated by the following correlation in Equation (20) [30]:

$$SD = \sqrt{\frac{1}{n-1} \sum_{k=1}^{n} \left( \frac{(\alpha_w)_{pred} - (\alpha_w)_{exP}}{(\alpha_w)_{exP}} \right)^2} \tag{20}$$

Figure 12 depicts a comparison of the predicted water holdup and the experimental data. The square, circular, and triangle symbols represent the data for $Fr_m = 0.105$, 0.216, and 0.368, respectively. The developed correlation (Equation (19)) demonstrates a good agreement with the experimental data. It is noted that the result measured by the water holdup instrument was smaller than the real water holdup $\alpha_w$ in the pipe when the input water cut $\varepsilon_w$ was less than 0.3. This was the reason for the predicted water holdup being away from the correlation reference line in the bottom-left part of the diagram. This discrepancy can be solved by improving the accuracy of the water holdup instrument in future research. Moreover, there was no evident structure to the degree of correlation with respect to the values of the Froude number $Fr_m$ of the oil–water mixture. This proved the applicability of Equation (19) for different Froude numbers. The power-law water holdup correlation is dimensionless, and it can be extended to other conditions. As can be seen in Figure 13, the classical Beggs-Brill empirical model [31] and the developed correlation (Equation (19)) in this study are used to predict the experimental water holdup data. The performance of Equation (19) is better than that of the Beggs-Brill model.

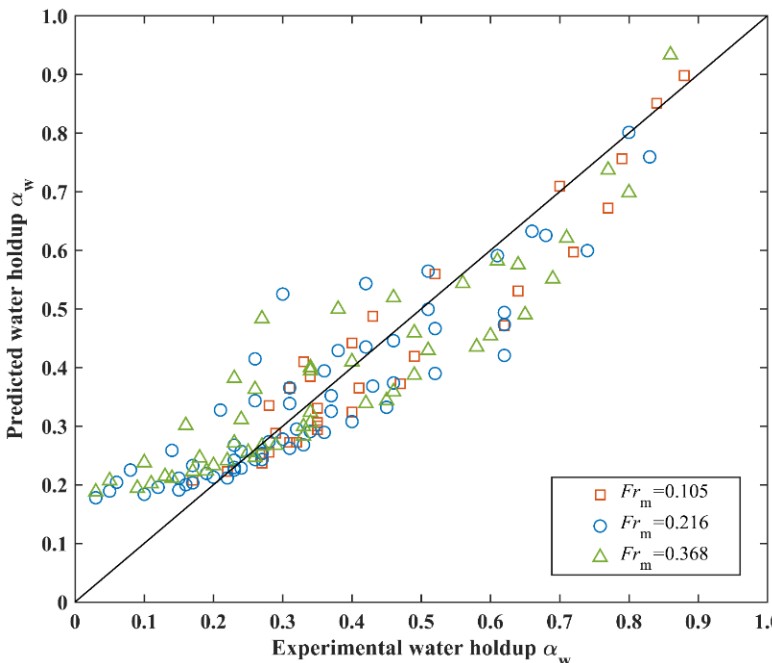

**Figure 12.** Comparison between the predicted water holdup and the experimental data for the oil–water–gas three-phase flow. The input water cut $\varepsilon_w$ ranged from 10 to 90%. The brown squares corresponded to $Fr_m = 0.105$, the blue circles corresponded to $Fr_m = 0.216$, and the green triangles corresponded to $Fr_m = 0.368$.

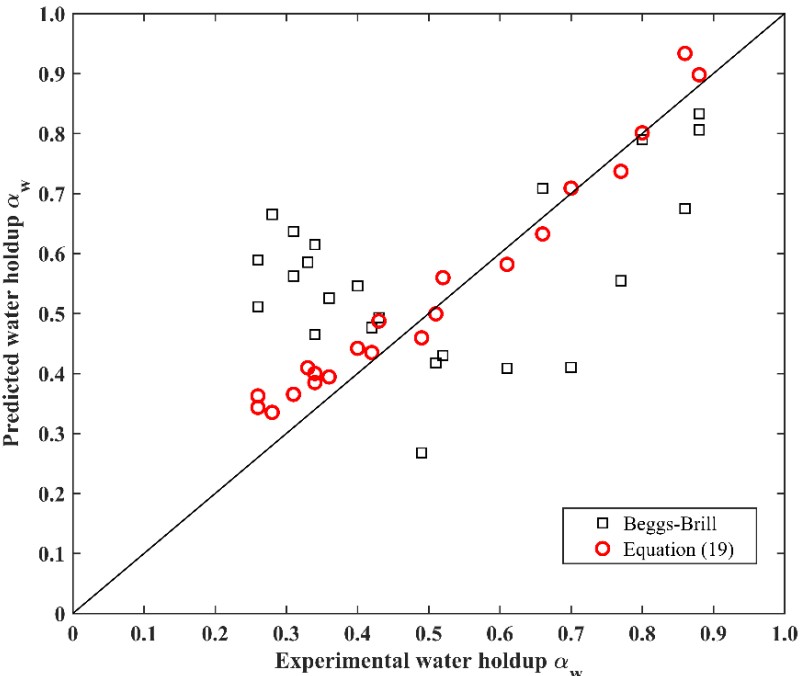

**Figure 13.** Comparison of different models predicting water holdup with the experimental data for the gas–water two-phase flow. The input water cut $\varepsilon_w$ is equal to 100%. The red circles corresponded to Equation (19), the black squares corresponded to the Beggs-Brill empirical model [31].

## 5. Conclusions

Due to the presence of water in oil wells or the injection of water into a well to increase the oil production, the pipe flow is in the form of oil–gas two-phase flows and oil–gas–water three-phase flows during oil exploitation and transportation. The change in the flow pattern and water holdup during oil pipeline transportation is important for the proper design and operation of pipelines. The change of the flow pattern has a significant effect on the pressure drop in the pipeline, and the calculation of the water holdup helps to predict the quantity of oil in petroleum pipelines. For example, the scaling and corrosion of a pipe can be prevented by controlling the proper flow pattern. In the long-distance multiphase transportation of high-viscosity crude oil, the flow can be artificially controlled as dispersed or annular flow with water as the continuous phase to reduce pressure drop losses. The judgment of the flow pattern and water holdup can also provide a quantitative basis for the separation of oil and water. Thus, a better understanding of the flow patterns and water holdup of the oil–water–gas three-phase flow is beneficial for the proper design and operation of pipelines. Meanwhile, it is also an essential topic in multiphase hydraulics and water resource management.

Oil–water–gas three-phase flow experiments were conducted in a pipe consisting of a horizontal section and a vertical section simultaneously. The newly designed water holdup instrument equipped with a conductance probe was used to measure the cross-sectional average water holdup. The flow behavior of the oil–water–gas three-phase flow was studied using dimensional analysis. The oil–water–gas three-phase flow was simplified as the two-phase flow of a gas and a liquid mixture. The effects of the dimensionless numbers $u_{sg}/u_{sm}$, $Re_m$, and $Fr_m$ on the multiphase flow in the pipe were analyzed and discussed.

New flow pattern maps of the three-phase flow in terms of $Re_m$ and $u_{sg}/u_{sm}$ were proposed over the range of superficial velocities considered. Plug, slug, and annular flows were observed in the horizontal section. Only slug and churn flows were observed in the vertical section. The plug flow was characterized by two peaks in the PDF, and the two peaks corresponded to the bubble region and the plug region, respectively. Slug flow was characterized by three peaks in the PDF, and the three peaks corresponded to the Taylor bubble, wake, and liquid slug regions, respectively. The annular and churn flows were

characterized by a single peak in the PDF, and the distribution range of the single peak in the churn flow was greater than that in the annular flow. Moreover, the flow pattern maps described using dimensionless numbers may also be applied to other pipe sizes. Based on the experimental data, a dimensionless power-law water holdup correlation for the oil–water–gas three-phase flow in the vertical section was developed. The predicted water holdup agreed reasonably well with the experimental results.

**Author Contributions:** Conceptualization: G.R., D.G. and P.L.; Formal analysis: G.R., D.G., P.L., X.Z. and X.L.; Investigation: G.R., P.L., X.Z. and X.L.; Methodology: G.R., X.C., K.S., R.F., L.M. and F.S.; Project administration: G.R., X.C., K.S., R.F., L.M. and F.S.; Supervision: G.R. and D.G.; Validation: P.L.; Visualization: P.L.; Writing—original draft: P.L.; Writing—review and editing: P.L., X.Z. and X.L. All authors have read and agreed to the published version of the manuscript.

**Funding:** This research received no external funding.

**Institutional Review Board Statement:** Not applicable.

**Informed Consent Statement:** Not applicable.

**Data Availability Statement:** Not applicable.

**Conflicts of Interest:** The authors declare no conflict of interest.

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
