# Peer review of "The Flow Pattern Transition and Water Holdup of Gas–Liquid Flow in the Horizontal and Vertical Sections of a Continuous Transportation Pipe"

_water, doi:10.3390/w13152077_

Round 1
Reviewer 1 Report
The manuscript presents experimental investigations of gas-liquid two phase flow, with the liquid being a water and oil mixture, in horizontal and vertical pipelines. The liquid content is measured by a conductance sensor. The authors measure the water holdup and observe the flow pattern for various water cuts. The results presented in this manuscript are original and can be interesting for the journal readers, although a journal on oil wells or fuel processing would be more relevant. However, there are many questions to be answered before the paper can be published, and I suggest major revision. English used in this manuscript is in general correct, but certain phrases, wording and grammar have to be edited and corrected by the authors.
Specific comments:
- Section 2. A couple of sentences should be given on the method of measurement by conductance sensor and electric equivalent scheme. It is not clear if the voltage drop across the sensor was measured by a constant current supply and how the voltage drop is distributed across the phases? Please provide more details on the method of measurement. From Figure 3 it can be supposed that the device measures the conductivity of water between central coaxial electrode and the pipe wall. But is it true? The electric conductivity of water depends also on the liquid temperature. It should be taken into consideration.
- Line 192. " The voltage values pertaining to the pure oil, water, and gas are denoted by Vo, Vw, and Vg, respectively.". This is a jargon statement. Must be corrected by the electrical-engineering phenomena.
- Line 199. "Vw as the characteristic voltage.. ". Probably the authors mean the voltage drop when the pipe is fully filled with water. It must be precisely explained.
- Line 206. "The voltage value of the mixture..". This does not sound physically. Please correct.
- Equation (8). The conductivity model of two-phase flow has to be more elaborated, with reference to the literature, in order to be convincing. This is not a simple problem of two or three dispersed phases, but the most often is used the model of two parallel electric capacitances (when a capacitance sensor is used) or parallel electric conductances. But this issue has not been solved analytically for more complex flow pattern since the first work by Bruggeman (Annalen der Physik 24, No.7, 636-679). Additionally it has to be noted that a method suitable to vertical channels cannot be used for horizontal channels and vv. This must be discussed thoroughly in Section 2. Without the explanation and knowledge of conductance sensor construction (cf. item 1) the reader cannot be sure if the proposed model is correct.
- Line 210. " voltage signals of the oil and gas are equal.." – jargon phrase – must be corrected in accordance with electrical phenomena.
- Line 213. " mean voltage of the mixture.." – the same.
- Line 216. " voltage values of pure water and pure oil.." – as above.
- Lines 407 and 414. "By analogy to.." is not correct. Please consider: "according to.. " or "from figure... results that..".
- Line 443. " The newly developed correlation (Eq. (19)) can predict the experimental data better. " – Please explain in what this correlation is better compared to another correlation (please specify this correlation)?
Author Response
Thanks for your thoughtful and constructive suggestions. We have carefully considered all the comments and revised the paper to the best of our abilities. Please find our responses listed below.

Reviewer 2 Report
In the manuscript Authors present the results of their own experiments on the water holdup in the water-oil-gas three-phase flow in the horizontal and vertical pipe sections. Before the article can be published, authors need to address some issues:
1) p.4, lines 155-159 - the water holdup instrument was placed 1 meter (20 pipe inside diameters) above the bottom of the vertical pipe section. I think that such a short pipe section may not be enough to reach the steady state. Did the Authors verified, that the flow structure does not change with the vertical pipe section length?
2) p. 5, lines 182-195 - I am not sure how the water holdup instrument used in the experiments functioned. Authors say, that "The conductance probe measured the voltage between the two ends of the conductor, i.e., the mean conductivity of the mixture in the pipe". Voltage drop is not conductivity. The conductivity was measured between the wire placed inside and the pipe wall? How exactly the conductivity was measured: constant current was flowing between two electrodes and the voltage drop was measured? AC or DC was used?
3) p. 7, lines 251-254 - viscosity of the liquid-liquid dispersions and emulsions may vary considerably with the water and oil cuts and may even be greater than the viscosity of pure liquids (in the vicinity of phase inversion point). In my opinion, Authors should use models that take into account the water share in the liquid-liquid mixture (see, e.g.: Hapanowicz, J., & Troniewski, L. (2002). Two-phase flow of liquid–liquid mixture in the range of the water droplet pattern. Chemical Engineering and Processing: Process Intensification, 41(2), 165-172).
4) p. 14, lines 385-390 - Authors were aware of the limitations of the water holdup instrument used in this study. Do the authors tried to estimate the uncertainty of the water holdup measurements?
5) p. 18, line 453 (Figure 12) - for the low water holdup predicted water holdups are much higher than experimental. Do the authors tried to use different type of correlation to predict the water holdup?
Author Response

(The authors gave the same response as above.)

Reviewer 3 Report
Dear authors the paper deals with an interesting industrial application giving an original contribution. The paper is almost ready for publication: the reviewer suggests to review the introduction section to include more relevant literature on the subject.
The reviewer suggest to include the following papers:
Du, M., Jin, N.D., Gao, Z.K., Wang, Z.Y. and Zhai, L.S., 2012. Flow pattern and water holdup measurements of vertical upward oil–water two-phase flow in small diameter pipes. International Journal of Multiphase Flow, 41, pp.91-105.
Leporini, M., Terenzi, A., Marchetti, B., Corvaro, F., Polonara, F., 2019, On the numerical simulation of sand transport in liquid and multiphase pipelines, Journal of Petroleum Science and Engineering, pp. 519-525, DOI: 10.1016/j.petrol.2018.12.057
Author Response

(The authors gave the same response as above.)

Round 2
Reviewer 1 Report
The authors have addressed all my comments and the paper can be published.
This manuscript is a resubmission of an earlier submission. The following is a list of the peer review reports and author responses from that submission.
Round 1
Reviewer 1 Report
The authors have modified the manuscript in accordance with the comments conducted by the reviewers. In my opinion, it can be accepted for publication in "Water".
Reviewer 2 Report
I would like to thank the authors for their kind reply to the comments and suggestions and for incorporating these in the revised manuscript.
After having read the revised manuscript, In the following, I provide a few additional comments and suggestions for improvement – hopefully constructive. Apologies for not including some of these already in the first review; some surfaced upon reading the revised manuscript.
Page 2, Before the Last paragraph - state (a) the objective, (b) the scope, (c) the contribution/added value of the present work/study. Then continue with the Layout of the paper.
Table 2, caption - “Data structure for the oil-water-gas pipe flow experiments.” The meaning of data structure is not clear. Would that mean “Parameter values of flow conditions imposed during the o-w-g pipe flow experiments”?
Table 2: Layout - I would suggest drawing a horizontal line for every cycle of input water-cut, εw, i.e. between lines 11 & 12 and 22 & 23. Also to align the values in the usg/usm column.
Figure 8, caption – I do not understand why flow pattern maps are presented in the same diagram for the horizontal and vertical sections of the test pipe. How can the reader understand if any flow pattern pertains to the horizontal or to the vertical cross-section? Or is it “just a typing error” in the figure caption?
Figure 12 - The degree of correlation between predicted and measured values of water hold-up is not discussed. It would be helpful to associate the discrepancy observed in the bottom left part of the diagram (framed) -whereby the markers are away from the correlation reference line, to the corresponding flow conditions. There is no obvious structure on the degree of correlation with respect to the values of the Froude number of the o-w mixture. It would be really interesting to read the opinion of the authors.
General comment 1, on the relevance of work - The authors should discuss the practical aspects of why/how the flow pattern and water hold-up value maps for different flow conditions (the object of the manuscript) are so important in the analysis and design of transport of gas and fluid mixtures.
Moreover, examples of practical applications of the present work into (any of) the following research areas (Water resources management; Water governance; Hydrology & hydraulics; Water scarcity; Flood risk; Water quality; Water & wastewater treatment; Urban water management; Water footprint assessment; Water-food; Water-energy; Water-human development; Water-ecosystems), would be especially welcome to justify thematic relevance to the Water MDPI Journal.
This comment is related to the previous comment, above, ref. “Page 2” or in the Conclusions section of the ms.
General comment 2, on text readability - Please provide a meticulous edit of the text (or ask for professional service); there are plenty of syntax and grammar errors that make it difficult to understand the text on a first pass/read. I have highlighted (in yellow) a few errors in the 1st section (Introduction).
Kind regards

Reviewer 3 Report
Page 3. All experiments are conducted using white mineral oil, distilled water, and air at atmospheric pressure and room temperature 20 Cº.
The pump may heat the fluid (water, oil). As the temperature changes, the fluid physical properties (viscosity, surface tension) may change. Fluid temperature should be monitored and controlled during the experiment.
Page 5. According to the mixture theory [2, 8], the voltage value of the mixture…
- Ibarra, R.; Nossen, J.; Tutkun, M., Two-phase gas-liquid flow in concentric and fully eccentric annuli. Part I: Flow patterns, holdup, slip ratio and pressure gradient. Chemical Engineering Science 2019, 203, 489-500.
- Spedding, P. L.; Donnelly, G. F.; Cole, J. S., Three phase oil-water-gas horizontal co-current flow. Chemical Engineering Research and Design 2005, 83, (4), 401-411.
In these papers, I found nothing about mixture theory and voltage.
Page 6. The oil-water-gas three-phase flow can be simplified as the two-phase flow of a gas and a liquid mixture
If the authors further consider two-phase flow, then the words three-phase flow in the paper title are inappropriate.
Page 6. The mixture viscosity is given as:
The assumption that the viscosity of a mixture is a linear superposition of the composing liquids is great error. Usually, it is more complex function. For example, viscosity of water-glycerin mixture has square-law dependence on concentration. It's a bad idea to treat a mixture of water and oil as a homogeneous fluid. Usually well mixed water and oil is emulsion. Unfortunately, the authors did not demonstrate a type of the mixture structure (emulsion, bubble flow, slug flow etc.). Moreover, usually oil is non-Newtonian liquid.
The same words about surface tension.
Page 10. The flow pattern maps have been developed…
Again, flow map depends strongly on channel size and size of dispersed phase inclusions. It will change with size variation.
